# Study of the Psychometric Properties of the Social Self-Efficacy Scale with Spanish Adolescents by Gender, Age and Family Socioeconomic Level

**DOI:** 10.3390/healthcare10061150

**Published:** 2022-06-20

**Authors:** Vanesa Salado, Diego Díaz-Milanés, Sara Luna, Sheila Velo

**Affiliations:** 1Department of Experimental Psychology, Faculty of Psychology, University of Seville, 41018 Seville, Spain; vsalado@us.es (V.S.); sluna@us.es (S.L.); mvelo@us.es (S.V.); 2Department of Psychology, Universidad Loyola Andalucía, 41704 Sevilla, Spain

**Keywords:** social self-efficacy, adolescents, invariance measurement, psychometrics, Spanish

## Abstract

Social self-efficacy has been shown to be a key resource for adolescents’ social experiences with peers and a predictor of prosocial behaviour among adolescents. However, differences by gender, age and socioeconomic level have previously been found in social self-efficacy. The objective of this study is to assess the psychometric properties of the subscale of social self-efficacy from the Self-Efficacy Questionnaire for Children (SEQ-C) developed by Muris (2001) in a representative sample of Spanish adolescents while considering gender, age and socioeconomic level differences. In general, the results showed good psychometric properties and a one-dimensional structure with high internal consistency, adequate explained variance and evidence of external validity for the subscale. Furthermore, the invariance analysis demonstrated that the social self-efficacy subscale shows no bias when used with populations of adolescents who differ by gender, age and socioeconomic level. The results indicate that the Spanish version of the social self-efficacy subscale of the SEQ-C is an adequate measurement instrument for assessing adolescents’ perception of their own social skills.

## 1. Introduction

A large number of studies have explored the general self-efficacy described by Bandura et al. [1] in adolescents [2,3,4], but only a few have focused on its specific dimensions, such as social self-efficacy [5,6], which can be defined as a young person’s ability to overcome social barriers [5] and develop healthy and supportive relationships [7].

Adolescence is a developmental stage during which young people are faced with many changes and influences from peers. These experiences highlight the need for the acute development of social self-efficacy that encourages the person to cope with adverse experiences with their peers and serves as protection against potential depressive or anxious symptoms [8]. On the other hand, the quality of relationships with friends and of lived experiences shared with them is a mediating factor between victimization and anxiety in young people [9].

High social self-efficacy in young people predicts prosocial behaviours for the benefit of their community [10,11,12] due to adolescents’ confidence in their own social competence [13]. Furthermore, boys and girls who trust in their social efficacy are more likely to build new and strong social relationships with their peers and are less likely to engage in disruptive behaviour [4].

From a gender perspective, some discrepancies have been found in the scores obtained by boys and girls in several studies. According to Coleman [14], 10–12-year-old girls show higher levels of social self-efficacy than boys. Gaspar et al. [15] found that after a social skills intervention in Portuguese schools, boys developed greater social skills than girls. This same study showed, from an evolutionary perspective, that as boys and girls grow older, they improve their socioemotional skills towards more complex levels, promoting greater social self-efficacy. Similarly, social support greatly enhances social self-efficacy in boys and girls [16]. Cicognani [17] asserted that older adolescents can adopt more effective strategies for finding solutions and resolving conflicts with others.

Regarding the socioeconomic level of the family, according to Bradley & Corwyn [18] and McLoyd [19], children from poorer homes face more economic challenges that impede their development and learning because their parents are unable to provide enough funds to support their cognitive stimulation (computers, trips, books, etc.). Along these lines, other studies indicated that potential learning problems in school were derived from the scarcity of resources and led to lower self-efficacy in all its dimensions [20,21]. However, Meilstrup et al. [22] showed that high self-efficacy and high social competence buffer the negative effects of low socioeconomic status on emotional symptoms among schoolchildren.

Due to the key role of social self-efficacy in different aspects of the development of adolescents, it is essential to consider measures of social self-efficacy that are supported by adequate evidence of reliability and validity for evaluations and interventions based on positive social interactions. In this regard, Muris [5] developed a questionnaire with the three dimensions of self-efficacy proposed by Bandura [1]—social, academic and emotional—in a sample of 330 young Europeans, which provided satisfactory results in internal consistency to the three subscales. Other researchers found similar results after adapting and assessing the psychometric properties of the instrument in their respective populations [23,24,25]. Among the three subdimensions, social self-efficacy measured by its 8-item version showed good construct validity and internal consistency in a sample of adolescents from the United States of America, with no differences by gender or age [6].

In view of the above, and given the relevance of social self-efficacy in the young population and its correlations with individual and contextual variables, the objective of this study is to assess the psychometric properties of the subscale developed by Muris [5], as Zullig et al. [6] did, in a representative sample of Spanish adolescents while attending to differences in gender, age and socioeconomic level.

## 2. Materials and Methods

### 2.1. Study Design and Participants

The study was based on a cross-sectional survey design. Population data were collected under the framework of the project the *Opinion Barometer of Childhood and Adolescence (Barometer OPINA)* by selecting a representative sample through random multistage sampling stratified by conglomerates.

The participants were 5773 adolescents from Spain: 3021 (52.3%) girls and 2752 (47.7%) boys. The age groups were distributed as follows: 1256 (21.8%) 11–12 years old, 2377 (41.2%) 13–14 years old, 1725 (29.9%) 15–16 years old and 415 (7.2%) 17–18 years old. The distribution by family socioeconomic level was 18.1% of adolescents with a low level, 45.6% with a medium level and 36.3% with a high level (Table 1). The young people responded to an online and anonymous questionnaire in their schools that respected their beliefs and privacy. Before the study was conducted, approval was granted by the *Comité de Ética de Andalucía (Andalusian Bioethics Committee)*.

### 2.2. Instruments

The present research is part of the Opinion Barometer of Childhood and Adolescence (Barómetro de Opinión de la Infancia y la Adolescencia) [26]. For this study, in addition to gender and age, the following instruments were included:*Family socioeconomic level* was evaluated through the latest version of the 6-item Family Affluence Scale (FAS-III) [27]. These items assess family material affluence through ownership of certain goods such as the number of cars, computers, or bathrooms. This instrument shows high test-retest reliability (r = 0.90) and consistency between child and parent reports (r = 0.80) [27]. In this research, FAS-III was employed as a categorical variable, distributing the subjects into three groups: the highest 20% classified as high-affluence, the lowest 20% as low-affluence, and the middle 60% as medium-affluence, as recommended in the last HBSC report [28].*Social self-efficacy* is an 8-item subscale from the Self-Efficacy Questionnaire for Children (SEQ-C) developed by Muris [5] and inspired by the general scale of self-efficacy of Bandura et al. [1]. The Spanish version was adapted through a translation/back-translation procedure carried out by two bilingual translators. Examples of items include “How well can you become friends with other children?”, “How well can you tell other children that they are doing something that you don’t like?”, “How well do you succeed in staying friends with other children?” and “How well do you succeed in preventing quarrels with other children?”. The response options ranged from 1 (not at all) to 5 (very well). Cronbach’s alpha was 0.85 for the original subscale [5]. For this study, the internal consistency was 0.82.*Perceived friend support* was evaluated through a subscale of the Multidimensional Scale of Perceived Social Support [29]. The items were “My friends really try to help me”, “I can count on my friends when things go wrong”, “I have friends with whom I can share my joys and sorrows” and “I can talk about my problems with my friends”. The response options ranged from 1 (very strongly disagree) to 7 (very strongly agree). The scale has been translated and adapted to different languages and population groups [30]. Both the original study [29] and studies based on samples of Spanish adolescents [31,32,33] have shown an adequate internal consistency of the perceived friend support subscale. The internal consistency of the subscale in the present study, estimated by Cronbach’s alpha, was α = 0.92.

### 2.3. Data Analysis

First, descriptive analyses were conducted to determine the mean and standard deviation of the social self-efficacy scale and its items, including the response percentage per item. Additionally, ceiling and floor effects were assessed for the total scale score. These effects were considered to be present if more than 15% of participants achieved the lowest or highest possible score, respectively [34,35]. Second, mean comparison analyses were carried out using Student’s *t* tests for independent samples to assess differences by gender in social self-efficacy and friends’ perceived support. The effect size of the differences was also obtained using Cohen’s d with the following cut-off points: 0 to 0.19 was considered negligible, 0.20 to 0.49 small, 0.50 to 0.79 medium and 0.80 and above high [36]. In addition, ANOVA was performed to compare social self-efficacy and friends’ support scores according to age and family socioeconomic level.

Third, internal consistency analyses were performed through Cronbach’s alpha, and the correlations between the score of each of the items and the total score of the scale were analysed for the global and segmented sample by gender, age and family socioeconomic level.

Furthermore, evidence of internal validity analysis referring to the structure of the questionnaire was assessed through confirmatory factor analysis using the unweighted least-squares method (ULS). Model fit was evaluated for both the global and segmented samples through different adjustment indices: chi-square (χ2), taking into account that it can be affected by sample size [37]; comparative fit index (CFI), with values above 0.90 considered acceptable; and root mean square error of approximation (RMSEA) and standardised root mean squared residual (SRMR), considering values near or below 0.08 and 0.05, respectively, as indicators of acceptable model fit. In addition, multigroup confirmatory factor analysis (CFA) was performed to evaluate configurational and metric invariance for the sample segmented by gender, age and family socioeconomic level, with an increase in CFI greater than 0.01 considered to be a significant change in the model [37].

Finally, to analyse the evidence of external validity referring to the relationship with other variables, the Pearson correlation coefficient was used to describe the association between social self-efficacy and friends’ support in the global and segmented samples.

A statistical significance of less than 5% (*p* < 0.05) was used for all cases. Statistical analyses were conducted using the statistical package IBM SPSS Statistics, version 26.0 (IBM Corp, Armonk, NY, USA), and JASP software, version 0.14 (JASP Team, Amsterdam, The Netherlands).

## 3. Results

### 3.1. Descriptive Statistics and Mean Comparison Analysis

Table 2 shows the descriptive analysis of the scale of social self-efficacy and the items that compose it. The number of participants with minimum (0.6%) and maximum (4.5%) scores on the social self-efficacy scale did not exceed the threshold for ceiling and floor effects. Table 3 presents the mean comparison analyses according to gender of social self-efficacy and the support of friends through Student’s *t* test. The results show that boys scored significantly higher on social self-efficacy than girls, with a negligible effect size. However, girls scored higher on friends’ perceived support, with a small effect size. Regarding age (social self-efficacy: F (3) = 0.611; *p* = 0.608; friends’ support: F (3) = 0.691; *p* = 0.557) and socioeconomic status (social self-efficacy: F (2) = 0.230; *p* = 0.795; friends’ support: F (2) = 0.313; *p* = 0.731), the results did not show significant differences for the variables studied.

### 3.2. Evidence of Reliability of the Questionnaire Scores

Cronbach’s alpha was 0.82 for the social self-efficacy scale, 0.83 for the sample of boys and 0.81 for the sample of girls, meeting the criteria defined a priori of an alpha greater than 0.7. The internal consistency was calculated by segmenting the sample into four age groups (11–12, 13–14, 15–16 and 17–18 years) and three socioeconomic levels (low, medium and high). Reliability indices ranged between 0.81 and 0.84, showing adequate internal consistency across all groups.

Correlations between the items and the social self-efficacy scale in the global sample were moderate and high, except for Item 8, for which the correlation was 0.27. In the sample of boys and girls, age and socioeconomic level correlations were similar to those in the global sample. Item 8 showed a correlation of 0.30 in the group of boys and 0.24 in the group of girls. Additionally, this item obtained correlations between 0.24 and 0.32 in the samples segmented by age and socioeconomic level, respectively.

In addition, reliability indicators showed that the model would improve if Item 8 were eliminated, obtaining a Cronbach’s alpha of 0.84 for the global sample. The elimination of this item would also produce a slight improvement in reliability in the sample segmented by the different age groups and socioeconomic levels, with differences of approximately 0.02 points in alpha.

### 3.3. Evidence of Internal Validity Regarding the Structure of the Questionnaire

The fit indices for the social self-efficacy model in the global sample were excellent (χ2/df = 36.08; NNFI = 0.981; CFI = 0.989; IFI = 0.989; RMSEA = 0.008; SRMS = 0.037). Even though Chi-square was significant, Cheung & Rensvold [37] observed that for large samples, Chi-square is sensitive, so it is necessary to keep in mind the results of the rest of the indicators.

Figure 1 shows the estimated global model with the factor loadings of each item. The variance was between 31% and 58% for most items, with high estimated coefficients except for Item 8 (B = 0.36; *p* < 0.001; β = 0.28), which obtained an explained variance of 8.1%.

When the sample was segmented by gender, the items for the group of boys showed an explained variance greater than 30% in most of the items, except in Item 8 (B = 0.41; *p* < 0.001; β = 0.32), which showed 10.2%. In the group of girls, this item was even lower, with an explained variance of 6.3% (B = 0.32; *p* < 0.001; β = 0.25). The rest of the items in this group had explained variances of between 30.7% and 61.3%.

In the age-segmented sample, the items for all groups explained between 30% and 65% of the variance. Furthermore, Item 8 did not exceed 10% in the 13–14, 15–16 and 17–18 years groups. In the 11–12 age group, this item showed an explained variance of 12.4% (B = 0.44; *p* < 0.001; β = 0.35). Similar results were obtained in the three socioeconomic level groups, with values of variance that were between 7.6% and 31.6%, with Item 8 showing values less than 10% (low: B = 0.35; *p* < 0.001; β = 0.27; middle: B = 0.35; *p* < 0.001; β = 0.28; high: B = 0.38; *p* < 0.001; β = 0.30).

Table 4 shows the results of the factorial invariance analysis by gender, age and socioeconomic level. First, configurational invariance showed an adequate fit of the model to the data for all samples. Subsequently, the metric invariance also showed an adequate adjustment, despite the restriction of parameters. There was no increase in CFI greater than 0.01, so the measurement invariance was confirmed in all subsamples.

### 3.4. Evidence of External Validity Regarding the Relationship with Other Variables

Pearson’s correlation coefficient values indicate positive, moderate and significant relationships (*p* < 0.01) between social self-efficacy and friends’ support (r = 0.41).

The sample segmented by gender showed positive and moderate relationships between variables, with the sample of boys having a greater correlation with self-efficacy (r = 0.46) than the sample of girls (r = 0.38).

Finally, the sample segmented by age obtained moderate and positive correlations in all groups (11–12 years: r = 0.43; 13–14 years: r = 0.39; 15–16 years: r = 0.43; 17–18 years: r = 0.39) as well as when divided into the three socioeconomic levels (low: r = 0.43; middle: r = 0.42; high: r = 0.39).

## 4. Discussion

To the best of our knowledge, this is the first study that explores the psychometric properties of the social self-efficacy subscale developed by Muris [5] in Spanish adolescents from a gender, evolutive development and socioeconomic perspective.

The social self-efficacy of young people enabled them to relate to others in an effective and healthy manner, avoiding conflicts and sharing experiences [38,39]. As a result of this instrument’s evaluation, we will be able to learn about the characteristics of adolescents to promote healthy emotional and social development.

In this regard, preliminary results of mean comparisons in social self-efficacy, taking into account various sociodemographic variables, revealed that boys have higher beliefs about their skills to develop healthy social relationships than girls. However, girls scored higher on perceived friend support, confirming the findings of Rueger et al. [40] and Shaheen et al. [41] on the importance of friendship in adolescent development. Regarding the social self-efficacy score, the results support the findings of Coleman [14], indicating a higher level of self-efficacy in boys from an early age. On the other hand, these results do not support other studies in which girls had greater self-efficacy than boys [6,42] or in those that did not find differences by gender [43,44]. Nonetheless, we must exercise caution when interpreting our results because the intensity of these differences indicates a negligible size effect, implying that statistically significant differences may be due to the large sample size. Regarding age and socioeconomic level, there do not seem to be differences in social self-efficacy and perceived social support provided by friends in any of the groups. The results do not support the findings of Cicognani [17], who reported higher scores in social self-efficacy in older adolescents, nor the works by Bradley & Corwyn [18] and McLoyd [19] on social inequalities in the development and learning of different skills.

From a psychometric approach, the subscale showed adequate reliability in our sample, as Muris [5] and Zullig et al. [6] found. Similar results were obtained when the sample was segmented by gender, age and socioeconomic level, showing good indicators of internal consistency for all the cases. Only removing Item 8 (“How well do you succeed in preventing quarrels with other children?”) increased the reliability, but the increment was negligible.

Regarding the evidence for validity, an analysis of the structure of the instrument in the total sample and every subsample revealed that the items showed factor loadings in a range between 0.30 and 0.80 with excellent goodness-of-fit for the model. Although Muris [5] decided to remove Item 8 with a loading of 0.32, this action should be interpreted with caution. It is possible that many adolescents have never experienced a situation that they could consider as a “quarrel” and so answered the question based on their perception and not on their experience. Additionally, the concept could be quite different between individuals who consider verbal confrontations and those who consider physical aggression to be indicators of a quarrel. Therefore, preserving this item could provide some additional information about how competent the respondents perceive themselves to be in such situations and what they consider to be a dispute. Future research should include a more detailed explanation about the concept “quarrel”. In addition, metric measurement invariance was tested for each sociodemographic variable, and under parameter constraint for each subsample, the goodness-of-fit indicators remained excellent, with factor loadings greater than 0.36 in all cases, including Item 8. Furthermore, no decrements greater than 0.01 in CFI were observed [37], which verified the model invariance and indicated that the instrument can be applied to all the groups analysed and their scores compared among them.

Finally, the external validity evidence regarding the relationship between social self-efficacy and the perceived social support provided by friends, both in the global sample and segmented by the different sociodemographic variables, showed a positive and moderate association in all groups.

It is necessary to highlight that a greater perceived ability to establish healthy social relationships among young people of any age and socioeconomic level is related to the search for emotional and social support from their peers and to the avoidance of stressful experiences such as bullying [45]. This context indicates that social self-efficacy can diminish and buffer situations that reduce their personal satisfaction and help them build relationships that increase their subjective wellbeing [46,47]. In addition, the promotion of adequate self-efficacy and social competence through school initiatives can cushion and improve the negative effects on minors’ mental health caused by socioeconomic inequalities [22].

This study has several limitations that must be acknowledged. One is that all the data obtained were collected through self-administered instruments. In addition, due to the study’s cross-sectional design, it is not possible to establish causal relationships between the variables studied, which would require a longitudinal study that would allow us to verify the directionality and causality of the relationships explored. Finally, the sample can be considered representative of Spanish adolescents, but the 17–18-year-old group may be biased because the data were collected in educational institutions, whereas attending school at this educational stage is not mandatory. However, as the main strength of the study, the results showed that the invariance of the instrument with differences in gender, age and socioeconomic level verifies the stability and applicability of this instrument in different contexts. Furthermore, the specificity and brevity of the instrument allow its incorporation into more complex issues or as the subject of interventions in various educational programs.

Future research may be able to assess the psychosocial properties of social self-efficacy in a variety of culturally diverse contexts, investigate which other specific subdimensions of general self-efficacy [1] are relevant in adolescent development and examine how to incorporate these findings into educational policies and mental health promotion initiatives for youth.

## 5. Conclusions

The social self-efficacy subscale from the Self-Efficacy Questionnaire for Children (SEQ-C) developed by Muris [5] showed good psychometric properties in a Spanish sample of adolescents. The results showed that the instrument had a one-dimensional structure with high internal consistency and an adequate explained variance.

Greater scores in social self-efficacy were related to a higher perceived level of social support from friends. According to previous research, higher social self-efficacy is associated with the development of skills to deal with adversity and to build safe and healthy relationships with peers. Therefore, this study provides new evidence of the external validity of the subscale. Furthermore, the subscale can be used in different populations despite differences in gender, age or socioeconomic level since the factor invariance allows us to compare their scores on the same metric. Additionally, a more detailed description of the concept of “quarrel” in Item 8 should be considered in future studies.

The results support that the instrument is a useful tool for developing preventive mental health programs in educational centres and allowing policy-makers to design tailored policies for adolescents.

## Figures and Tables

**Figure 1 healthcare-10-01150-f001:**
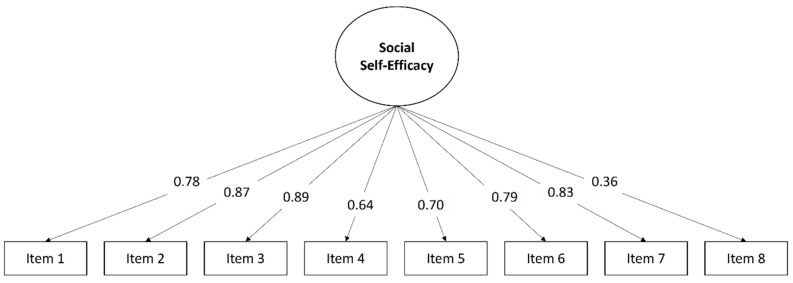
Estimates coefficients of items in the global model.

**Table 1 healthcare-10-01150-t001:** Sample description.

Demographic	Value	Frequency	%
Participants	N° of Participants	5773	100%
Gender	Boys	2752	47.67%
	Girls	3021	52.33%
Age group (years)	11–12	624	22.67%
	13–14	1081	39.28%
	15–16	860	31.25%
	17–18	187	6.80%
Family socioeconomic level	Low	478	18.31%
	Middle	1184	45.36%
	High	948	36.32%

**Table 2 healthcare-10-01150-t002:** Descriptive analysis of each item and the global social self-efficacy scale (*n* = 5773 adolescents, 13–18 years old).

Items	Mean	*SD*	1	2	3	4	5
Social self-efficacy score	3.62	0.80					
Item 1. How well can you express your opinions when other classmates disagree with you?	3.48	1.27	11.2%	9.1%	25%	29%	25.7%
Item 2. How well can you become friends with other children?	3.78	1.17	6.1%	8.6%	19.6%	32%	33.7%
Item 3. How well can you have a chat with an unfamiliar person?	3.32	1.27	11.7%	13.8%	26.3%	26.5%	21.7%
Item 4. How well can you work in harmony with your classmates?	3.72	1.07	4.9%	7.8%	23.4%	38.2%	25.7%
Item 5. How well can you tell other children that they are doing something that you don’t like?	3.46	1.26	9.4%	13.2%	24.7%	27.2%	25.6%
Item 6. How well can you tell a funny event to a group of children?	3.64	1.22	7.7%	10%	22.7%	29.4%	30.2%
Item 7. How well do you succeed in staying friends with other children?	3.84	1.08	4.3%	6.8%	22%	34.2%	32.7%
Item 8. How well do you succeed in preventing quarrels with other children?	3.66	1.27	8.9%	9.9%	20.3%	27.3%	33.7%

Note: *SD*: standard deviation; Numbers from 1–5 indicates the response option where 1 means “not at all” and 5 means “very well”.

**Table 3 healthcare-10-01150-t003:** Means comparison analysis of variables by gender.

Variables	Descriptive Statistics				Significance Tests and Effect Size
	$\bar{x}$	*SD*	$\bar{x}$	*SD*	
	Boys		Girls		
Social self-efficacy	3.64	0.80	3.59	0.80	*t* _(5771)_ = 2.31, *p* = 0.021; *d* = 0.06
Friends support	5.53	1.43	5.93	1.34	*t* _(5629)_ = −11.04, *p* < 0.001; *d* = 0.28

Note: $\bar{x}$: mean; *SD*: standard deviation; *t*: Student’s t; *d*: Cohen’s d.

**Table 4 healthcare-10-01150-t004:** Goodness of fit indices for the different steps of the factorial invariance analysis.

Models		χ^2 a^/gl ^b^	NNFI ^c^	CFI ^d^	∇ CFI ^e^	IFI ^f^	RMSA ^g^ (CI 95%) ^g^	SRMS ^h^
Configurational invariance	Gender	21.29	0.978	0.987	0.002	0.987	0.008	0.039
	Age	9.08	0.982	0.987	0.002	0.987	0.007	0.041
	FAS ^i^	10.82	0.983	0.988	0.001	0.988	0.007	0.040
Metric invariance	Gender	19.71	0.979	0.986	0.003	0.986	0.008	0.004
	Age	8.70	0.983	0.985	0.004	0.985	0.073	0.045
	FAS	9.13	0.986	0.987	0.002	0.987	0.067	0.041

^a^ χ2, Chi squared; ^b^ df, degree of freedom; ^c^ NNFI, non-normed fit index; ^d^ CFI, comparative fit index; ^e^ ∇ CFI, decrease in CFI; ^f^ IFI, incremental fit index; ^g^ RMSA, root mean squared error; ^h^ SRMR, standardised root mean squared residual; ^i^ FAS, Family Affluence Scale.

## Data Availability

The data that support the findings of this study are available upon request from the authors.

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
