# Peer review of "Study of the Psychometric Properties of the Social Self-Efficacy Scale with Spanish Adolescents by Gender, Age and Family Socioeconomic Level"

_healthcare, 2022, doi:10.3390/healthcare10061150_

Round 1

Reviewer 1 Report

The research described in this manuscript appears to have been conducted properly; however, the manuscript as written is difficult to understand due to the frequency of mistakes in English usage and style. There are numerous errors as well as questionable wording in the abstract, including but not limited to the following: 

  • Lines 10-12 (lack of clarity) - The first sentence should be rewritten because it is not clear what you are trying to say. Is social self-efficacy both a key resource for adolescents' social experiences with peers AND a predictor of prosocial behavior (among adolescents)? Also, what is meant by "being able to find some differences by gender, age and socioeconomic level"? Who is "being able to find some differences"?
  • Line 13 - (incorrect wording) Is "form" the right word?
  • Lines 14-15 (misplaced modifier) - Not a "Spanish sample of children and adolescents", but rather a "sample of Spanish children and adolescents". The same error appears in line 71 on p. 2.
  • Lines 15 & 18 (inconsistency in comma usage) - Compare line 15 - "gender, age, and socioeconomic level" with line 18 - "gender, age and socioeconomic level". Inconsistency in the use of the Oxford comma appears throughout the manuscript, including in the title. 
  • Lines 15-17 (lack of clarity) - "In general, results showed good psychometric properties and a one-dimensional structure with high internal consistency, adequate explained variance and external validity for the subscale." This sentence should be rewritten. Did you mean that your results showed a one-dimensional structure, high internal consistency, adequate explained variance, and evidence of external validity for the subscale
  • Line 17 (incorrect wording) - "Besides" is not used correctly in this context, which is repeated in lines 199 and 311.
  • Lines 17-19 (questionable claim) - The sentence, "The invariance analysis made by gender, age and socioeconomic level showed that it can be used in different populations with no bias appearance" could be an overstatement. I would argue that the invariance analysis demonstrated that the social self-efficacy subscale shows no bias when used with populations of adolescents who differ by gender, age, and socioeconomic level.
  • Lines 19-20 - This is another sentence that should be rewritten, for example, "The results indicate that the Spanish version of the social self-efficacy subscale of the SEQ-C is an adequate measurement instrument for assessing adolescents' perception of their own social skills."

Besides improving the use of English throughout the manuscript, there are additional problems:

  • Line 29 - "Childhood, adolescence, and youth are developmental stages" infers that each of these (i.e., childhood, adolescence, and youth) are separate stages, when in fact they are not mutually exclusive. There is no universal agreement when childhood ends and adolescence begins. The term "youth" can apply to children, adolescents, and even young adults.
  • Lines 49-51 - I have three concerns about this sentence: "Regarding the socioeconomic level of the family, no research has found a causal relationship between a lower socioeconomic level and lower social self-efficacy, this might be due to the descriptive nature of the variable Family Affluence Scale [17]." (1) Unless all of the research that examined the relationship between family socioeconomic level and self-efficacy utilized the Family Affluence Scale, you cannot make the claim that "no research has found a causal relationship ... due to the descriptive nature of the Family Affluence Scale." (2) Is there any research that found a correlation between the two variables? (3) What do you mean by "the variable Family Affluence Scale"? Is the scale itself variable? 
  • Line 91 - Did you use a Spanish version of the revised Family Affluence Scale (FAS III)? If so, how was it translated? How many items are on the FAS III? What are the psychometric properties of the English and the Spanish versions of the FAS III?
  • Lines 92-93 - Although you mentioned the Self-Efficacy Questionnaire for Children in the abstract and introduced its acronym (SEQ-C), the SEQ-C was never mentioned again (except in the References). Readers might want to know what the parent instrument for the social self-efficacy subscale is. You should also identify the parent instrument on line 303.
  • Lines 102-106 - Were there only four items on the subscale you used to measure perceived friend support?
  • Lines 107-109 - Your statement that the translated versions of the Multidimensional Scale of Perceived Social Support show "in general, adequate psychometric properties" is not true. Please refer to https://www.ncbi.nlm.nih.gov/pmc/articles/PMC5930820/, especially Table 2 and the paragraph that describes the psychometric properties of the Spanish version of the MSPSS. 
  • Lines 109-110 - "The internal consistency of the instrument in the present study, estimated by Cronbach's alpha, was α = 0.92." Was the entire MSPSS administered to the sample? It would be incredible for a four-item subscale to have that level of internal consistency.
  • Line 184 - This appears to be a subtitle (Subsection 3.3?). If so, then Evidence of external validity referred to the relationship with other variables on line 217 should be Subsection 3.4.

Except for a few errors in English usage, the Discussion and Conclusion sections are well-written. Regardless, the manuscript requires major revisions and a more careful review before it is resubmitted.

Reviewer 2 Report

Based on the scale data of 5773 Spanish adolescents, the current study aimed to assess the psychometric properties of the subscale of social self-efficacy while attending to gender, age, and socioeconomic level differences. The results showed good psychometric properties and a one-dimensional structure with high internal consistency, adequate explained variance and external validity for the subscale. Moreover, the invariance analysis made by gender, age and socioeconomic level showed that it can be used in different populations with no bias appearance. Overall, this is an interesting and potentially important piece of the research. However, there are a number of concerns over that preclude publication at this time.

1) The description of the subjects needs to be consistent in the study. For example, Spanish Adolescents in the title, Spanish sample of children and adolescents in the Abstract section, “Participants were 5773 adolescents from Spain” in the Study design and participants section.

2) Except for Social self-efficacy, an 8-item subscale general scale of self-efficacy, the authors also measured the Family Affluence Scale and Multidimensional Scale of Perceived Social Support of participants. However, I did not find the theoretical and rational reasons why they measured the Family Affluence Scale and Multidimensional Scale of Perceived Social Support. It would be more clear if the authors added some theoretical description of the reason to the introduction section.

3) I appreciated the detailed description of the participants in the “Study design and participants”. It would be more clear if the authors added the demographic information, e.g., age, gender, family socio-economic level and so on, to a table.

4) Font size is inconsistent in sentence “On the other hand, Table 2 presents the mean comparison analyses according to gender of social self-efficacy and the support of friends through Student’s t test.”

Reviewer 3 Report

The manuscript addresses the study of the Psychometric Properties of Social Self-Efficacy Scale with Spanish adolescents by gender, age and family.

Social self-efficacy has shown to be a key resource to cope with social experiences with peers in adolescence and a predictor of prosocial behavior for the community, being able to find some differences by gender, age and socioeconomic level. The objective of this study is to assess the psychometric properties of the subscale of social self-efficacy form the Self-Efficacy Questionnaire for Children (SEQ-C) developed by Muris (2001), in a representative Spanish sample of children and adolescents while attending to gender, age, and socioeconomic level differences. In general, results showed good psychometric properties and a one-dimensional structure with high internal consistency, adequate explained variance and external validity for the subscale. Besides, the invariance analysis made by gender, age and socioeconomic level showed that it can be used in different populations with no bias appearance. All this makes the Spanish version of the social self-efficacy sub-scale an adequate measurement to assess the perception of own social skills in adolescence. 

First of all, thank you for the possibility to review the manuscript,

It's a good job, on a very interesting topic such as self-efficacy, but before publication should incorporate the following suggestions, I ask the authors to follow them one by one, 

in the introduction some important references are missing, revise it,

the explanation of the procedure carried out for the investigation needs to be improved, 

the discussion section must be improved and more worked, much more worked, so that a work can be published in our journal,

The bibliography of the previous studies of the introduction must be connected with the discussion, that will give more power to the manuscript,

Finally, the bibliography should be ordered, following the rules of the journal.

with these changes made, the article will improve and will have the quality to be published in this prestigious journal

kind regards

Round 2

Reviewer 1 Report

This manuscript is much improved from the previous one. I noted a few minor errors that should be corrected:

Line 19 - delete serial comma (aka Oxford comma) after "age"

Line 48 - change "assert" to "asserted"

Line 65 - change "in" to "on"

Line 107 - delete serial comma after "computers"

Reviewer 2 Report

The authors have addressed all of my concerns